

# T cell mediated immunity against influenza H5N1 nucleoprotein, matrix and hemagglutinin derived epitopes in H5N1 survivors and non-H5N1 subjects

Pirom Noisumdaeng[1,2,3], Thaneeya Roytrakul[4], Jarunee Prasertsopon[5], Phisanu Pooruk[6], Hatairat Lerdsamran[5], Susan Assanasen[7], Rungrueng Kitphati[8], Prasert Auewarakul[3] and Pilaipan Puthavathana[3,5]

[1] Faculty of Public Health, Thammasat University, Khlong Luang, Pathum Thani, Thailand
[2] Thammasat University Research Unit in Modern Microbiology and Public Health Genomics, Thammasat University, Khlong Luang, Pathum Thani, Thailand
[3] Department of Microbiology, Faculty of Medicine Siriraj Hospital, Mahidol University, Bangkok-noi, Bangkok, Thailand
[4] National Center for Genetic Engineering and Biotechnology, Khlong Luang, Pathum Thani, Thailand
[5] Center for Research and Innovation, Faculty of Medical Technology, Mahidol University, Nakhon Pathom, Thailand
[6] The Government Pharmaceutical Organization, Biological Product Vaccine Production Plant, Kaengkhoi, Saraburi, Thailand
[7] Department of Medicine, Faculty of Medicine Siriraj Hospital, Mahidol University, Bangkok-noi, Bangkok, Thailand
[8] Ministry of Public Health, Nonthaburi, Thailand

Corresponding author
Pilaipan Puthavathana,
pilaipan.put@mahidol.edu

## ABSTRACT

**Background.** Protection against the influenza virus by a specific antibody is relatively strain specific; meanwhile broader immunity may be conferred by cell-mediated immune response to the conserved epitopes across influenza virus subtypes. A universal broad-spectrum influenza vaccine which confronts not only seasonal influenza virus, but also avian influenza H5N1 virus is promising.

**Methods.** This study determined the specific and cross-reactive T cell responses against the highly pathogenic avian influenza A (H5N1) virus in four survivors and 33 non-H5N1 subjects including 10 H3N2 patients and 23 healthy individuals. Ex vivo IFN-$\gamma$ ELISpot assay using overlapping peptides spanning the entire nucleoprotein (NP), matrix (M) and hemagglutinin (HA) derived from A/Thailand/1(KAN-1)/2004 (H5N1) virus was employed in adjunct with flow cytometry for determining T cell functions. Microneutralization (microNT) assay was performed to determine the status of previous H5N1 virus infection.

**Results.** IFN-$\gamma$ ELISpot assay demonstrated that survivors nos. 1 and 2 had markedly higher T cell responses against H5N1 NP, M and HA epitopes than survivors nos. 3 and 4; and the magnitude of T cell responses against NP were higher than that of M and HA. Durability of the immunoreactivity persisted for as long as four years after disease onset. Upon stimulation by NP in IFN-$\gamma$ ELISpot assay, 60% of H3N2 patients and 39% of healthy subjects exhibited a cross-reactive T cell response. The higher frequency and magnitude of responses in H3N2 patients may be due to blood collection at the convalescent phase of the patients. In H5N1 survivors, the effector peptide-specific T cells generated from bulk culture PBMCs by in vitro stimulation

displayed a polyfunction by simultaneously producing IFN-$\gamma$ and TNF-$\alpha$, together with upregulation of CD107a in recognition of the target cells pulsed with peptide or infected with rVac-NP virus as investigated by flow cytometry.

**Conclusions**. This study provides an insight into the better understanding on the homosubtypic and heterosubtypic T cell-mediated immune responses in H5N1 survivors and non-H5N1 subjects. NP is an immunodominant target of cross-recognition owing to its high conservancy. Therefore, the development of vaccine targeting the conserved NP may be a novel strategy for influenza vaccine design.

# INTRODUCTION

Influenza A virus infection is common in a wide range of avian and mammalian hosts (*Webster et al., 1992*; *Horimoto & Kawaoka, 2005*; *Taubenberger & Kash, 2010*; *Long et al., 2019*). Some avian influenza A virus subtypes, i.e., H5N1, H5N6, H7N7, H7N9 and H9N2, are able to cross the species barrier and infect humans (*Wong & Yuen, 2006*; *Writing Committee of the Second World Health Organization Consultation on Clinical Aspects of Human Infection with Avian Influenza A, H5N1)Virus(2008*; *Yu et al., 2013*; *Puzelli et al., 2014*; *Pan et al., 2016*; *Peacock et al., 2019*). Among these avian influenza subtypes, the H5N1 highly pathogenic avian influenza (HPAI) virus is the most virulent and the most fatal in humans and animals (*Peiris, Jong & Guan, 2007*; *Korteweg & Gu, 2008*). The first outbreak of the H5N1 HPAI virus in humans occurred in Hong Kong in 1997. Its re-emergence in 2003 affected several countries in various continents, in particular, the Asia-Pacific region (*Chan, 2002*; *Horimoto & Kawaoka, 2005*; *Peiris, Jong & Guan, 2007*). From 2003 to 10 July 2020, the World Health Organization (WHO) reported a cumulative number of 861 human cases with 455 deaths leading to the fatality rate of approximately 53% (*World Health Organization, 2020*). A person infected with the H5N1 HPAI virus mostly exhibits severe pneumonia, while asymptomatic infection or mild illness was scant (*Hinjoy et al., 2008*; *Le et al., 2013*). Almost of the cases contracted the H5N1 virus infection through contact with sick or dead poultry. Rare cases of person-to-person transmission have been documented (*Ungchusak et al., 2005*). Interestingly, nearly all villagers living in the H5N1 outbreak areas are naïve to this virus infection (*Hinjoy et al., 2008*; *Dejpichai et al., 2009*), which anticipated that their immune system is naïve to this virus subtype.

There were several studies on the immune responses to H5N1 virus infection based on the in vitro or ex vivo system of animal origins (*O'Neill et al., 2000*; *Seo & Webster, 2001*; *Seo, Peiris & Webster, 2002*; *Droebner et al., 2008*; *Sawai et al., 2008*; *Galli et al., 2009*; *Richards, Chaves & Sant, 2009*; *Van Maurik et al., 2010*; *Rimmelzwaan & Katz, 2013*; *Lin et al., 2013*; *Ross et al., 2014*; *Park et al., 2014*; *Koutsakos, Kedzierska & Subbarao, 2019*). Conversely, there is limited information on the immune responses against H5N1 virus

infection in humans, particularly in terms of cell-mediated immunity (CMI) which plays an essential role on viral clearance by eliminating the virus-infected cells (*Boon et al., 2004*; *Thomas et al., 2006*; *Mbawuike, Zhang & Couch, 2007*; *Rimmelzwaan & Katz, 2013*; *Koutsakos, Kedzierska & Subbarao, 2019*). Several studies demonstrated that a majority of T cells specific to the seasonal H3N2 or H1N1 influenza virus recognize a variety of the viral proteins: nucleoprotein (NP), matrix (M), polymerase (PB1, PB2, PA), hemagglutinin (HA), neuraminidase (NA) and nonstructural (NS) proteins (*Assarsson et al., 2008*; *Gioia et al., 2008*; *Kreijtz et al., 2008*; *Lee et al., 2008*; *Roti et al., 2008*; *Babon et al., 2009*). The CD4$^+$ and CD8$^+$ T cells which targeted the seasonal influenza viral proteins also cross-recognized the internal conserved epitopes in H5N1 proteins: PB1, PB2, PA, NS, and in particular, the NP and M which are the immunodominant targets (*Assarsson et al., 2008*; *Gioia et al., 2008*; *Kreijtz et al., 2008*; *Lee et al., 2008*; *Roti et al., 2008*; *Babon et al., 2009*). Furthermore, the cross-reactive memory CD4$^+$ T cells recognized the variable surface glycoprotein HA and NA (*Roti et al., 2008*). Nevertheless, this information was generated by using immune cells from healthy donors due to an inability to access the blood specimens from H5N1 patients. Therefore, the information obtained may not represent the immune response developed after H5N1 natural infection.

In this study, we assessed the T cell response to H5N1 NP, M, and HA epitopes using archival peripheral blood mononuclear cells (PBMCs) from individuals who recovered from H5N1 HPAI viral infections from 2004 to 2005 using flow cytometry and IFN-$\gamma$ ELISpot assay. Moreover, we investigated the cross-reactive T cell response to H5N1 proteins in individuals who recovered from the H3N2 influenza virus infection. Our study may aid the design of a candidate T cell-based universal vaccine for broad-viral subtype protection.

## MATERIALS AND METHODS

### Ethical issue

This study was approved by Institutional Review Boards from the Faculty of Medicine Siriraj Hospital, Mahidol University, under approval number Si213/2005. Written informed consent was obtained from all non-H5N1 individuals and H5N1 survivors or their parents for participation in this study (*Kitphati et al., 2009*; *Noisumdaeng et al., 2014*).

### Human subjects and blood specimen

Sera, plasma and peripheral blood mononuclear cells (PBMCs) were obtained from 37 participants including four H5N1 Thai survivors, 10 H3N2 patients and 23 healthy individuals (*Kitphati et al., 2009*; *Noisumdaeng et al., 2013*; *Noisumdaeng et al., 2014*). Survivor nos. 1 and 2 were adults, while survivor nos. 3 and 4 were young children. All of them were infected with H5N1 clade 1 virus. A total of 20 sequential blood samples were collected from these survivors at intervals for up to four years after disease onset (*Kitphati et al., 2009*). The demographic data of the H5N1 survivors and time at blood specimen collection are shown in supplementary Table S1A. The H3N2 patients were diagnosed by real time reverse transcription-polymerase chain reaction (real time RT-PCR), and virus isolation together with serodiagnosis when possible. Demographic data of non-H5N1

infected subjects is presented in supplementary Table S1B. Sera and plasma samples were kept frozen at −20 °C until used. PBMCs were separated from anti-coagulated blood using Ficoll-Hypaque, (IsoPrep, Robbins Scientific Corporation, Sunnyvale, CA) density gradient centrifugation and stored in a freezing medium containing 10% DMSO (Sigma, MO) in fetal bovine serum (FBS) (Gibco®, NY) and cryopreserved in liquid nitrogen.

## Recombinant vaccinia viruses

Recombinant vaccinia virus carrying H5N1 NP gene inserted in the pSC11 plasmid backbone (rVac-NP virus) or the recombinant virus carrying only the pSC11 backbone (rVac-pSC11 virus) were constructed in our laboratory. The viruses were propagated and titrated in Thymidine kinase negative (TK⁻) cells maintained in Dulbecco's modified eagle medium (DMEM) (Gibco) supplemented with 10% FBS plus penicillin and streptomycin (*Noisumdaeng et al., 2013*; *Noisumdaeng et al., 2014*). The virus stocks were kept at −80 °C until used.

## Microneutralization (microNT) assay for antibody detection

ELISA-based microNT assay was carried out for detecting neutralizing antibodies against the A/Thailand/1(KAN-1)/2004 (H5N1) clade 1 (KAN-1 virus), A/Thailand/Siriraj-Rama-TT/2004 [A/New Caledonia/20/1999 (H1N1)-like virus], and A/Siriraj ICRC/SI-154/2008 [A/Brisbane/10/2007 (H3N2)-like virus]. The assay protocol was described previously (*Kitphati et al., 2009*; *Lerdsamran et al., 2011*; *Noisumdaeng et al., 2014*).

## Designing the overlapping peptides and peptide-pool matrix

Overlapping peptides spanning the entire NP, M1, M2 and HA proteins of H5N1 KAN-1 virus (GenBank accession no., AAV35112, AAV35110, AAV35111 and AAS65615, respectively) were synthesized in the PEPscreen®custom peptide libraries format (Sigma-Genosys, Singapore). According to supplementary Table S2, the amino acid sequence identities of each protein were conserved among various H5N1 viruses circulating in Thailand during the study period. According to the manufacturer, all peptides were analyzed by MALDI-TOF mass spectrometry and their average crude purity is greater than 70%. Each peptide was 20 amino acids long with 10 amino acid residues overlapping, except the last peptide of the protein which may be shorter. There were a total of 49 NP peptides, 25 M1 peptides, 10 M2 peptides and 56 HA peptides as shown in Tables S3A–S3C. Peptide powder was dissolved in dimethyl sulfoxide (DMSO, Sigma) to the concentration of 50 mg/ml, then aliquot and kept at −80 °C . The concentrate peptide solution was further diluted in serum-free Roswell Park Memorial Institute 1640 medium (RPMI, Gibco) to the concentration of 2.5 mg/ml and stored at −80 °C in aliquots. These peptides were mixed into individual pools according to the two-dimensional matrix system as shown in Tables S3D–S3F. Each peptide pool contained an individual peptide at working concentration of 40 μg/ml. There were a total of 14 pools (A1-A7 and B1-B7) for NP, 12 pools (A1-A6 and B1-B6) for M and 15 pools (A1-A8 and B1-B7) for HA. The DMSO concentration in each assay was less than 0.1% (v/v).

### Ex vivo IFN-γ enzyme-linked immunospot (ELISpot) assay

IFN-γ ELISpot assay was performed to demonstrate the T cell responses against H5N1-derived peptides among H5N1 survivors and non-H5N1 subjects. A 96-well polyvinylidene difluoride (PVDF) ELISPOT plate (Multiscreen[TM] IP, MAIPS4510, Millipore, USA) was coated with mouse monoclonal anti-human IFN-γ 1-D1K (Mabtech AB, Stockholm, Sweden) at a concentration of 15 μg/ml overnight at 4 °C, followed by blocking with RPMI supplemented with 10% FBS for 2 h. After the blocking solution was discarded, the PBMC suspensions were added at the concentration of $3 \times 10^5$ cells/100 μl/well. Thereafter, peptide pool at a final concentration of 4 μg/ml (for screening of T cell activity) or individual peptide (for peptide specific activity of T cells) at a final concentration of 10 μg/ml was added into each well. PBMCs incubated with medium only served as the negative control, whereas PBMCs treated with 1 and 10 μg of phytohemagglutitin (PHA) (Sigma) served as the positive controls. After incubation for 14–16 h, PBMCs were removed, and the plate was washed with 0.05% Tween-20 in PBS (PBST). The biotin conjugated-anti-IFN-γ (7-B6-1-biotin, Mabtech) was used as the primary antibody to bind the released IFN-γ absorbed on the membrane lining the plate bottom. Streptavidin-labelling alkaline phosphatase was used as the secondary antibody and followed by BCIP/NBT plus (Mabtech) as the chromogenic substrate. The reaction plate was incubated until purple spots were visible on the membrane, and then the reaction was terminated. The spots which represent the secreted IFN-γ producing T cells were counted and analyzed using a KS ELISpot version 4.11 program of an ELISpot plate reader (KS ELISpot, Zeiss, Munich, Germany). The T cell activity was considered positive when the number of spot forming cells (SFCs) in the reaction wells was at least two times greater than those of the negative control wells.

### Functional assays of H5N1-specific T cells

The T cell functional assay was demonstrated by intracellular cytokine staining (ICS) and flow cytometry of peptide-specific polyclonal T cell lines (effector T cells) which were generated as bulk culture by stimulating the PBMCs with an individual specific peptide. The study design was shown in Fig. S1, and the detailed method was as followed:

#### *Establishment of Epstein - Barr virus (EBV) transformed B cell lines (TBCLs)*

Autologous TBCLs were used as the target cells for T cell functional assay by flow cytometry. TBCLs were established by infecting PBMCs with EBV from supernatant of B95-8 culture in RPMI supplemented with 10% FBS and in the presence of cyclosprorin A. Clumps of transformed B lymphocytes generally appeared after 2–3 weeks of infection. TBCLs could multiply indefinitely, a condition known as immortalization.

#### *Establishment and maintenance of peptide-specific T cell lines*

A bulk culture of specific polyclonal T cell line was generated by stimulating PBMCs with peptide and used as a source of the effector T cells. Frozen PBMCs of an number of $5\text{-}10 \times 10^6$ cells were thawed and divided into two portions comprising 20% and 80% of total cells. The 20% portion were pulsed with 100 μg/ml of peptide before mixing with the 80% portion. The suspension of cell mixture was added into a 6-well culture plate at

the amount of $2-3 \times 10^6$ cells/well in 10% NHS RPMI supplemented with 25 ng/ml IL-7 (Peprotec, USA) and 50 U/ml IL-2 (Chiron, Netherlands).

Clumps of proliferating T cells should be seen in the culture at approximately 2 weeks after cultivation, i.e., peptide-specific polyclonal T cell line was established. T cell lines were expanded by stimulation with autologous gamma irradiated peptide-pulsed TBCLs plus irradiated allogeneic PBMCs (2,000 rad using cesium source).

### Target cell preparation

The peptide-pulsed autologous TBCLs or TBCLs infected with rVac-NP virus were used as the target cells or antigen presenting cells (APCs) for investigating the function of peptide-specific T cells by flow cytometry. The unpulsed TBCLs or TBCLs infected with rVac-pSC11 virus were used as the negative target cell control. Pellets of TBCLs were pulsed with 100 μg/ml of specific peptides for 1 h, followed by washing and re-suspending with 10% FBS RPMI. For preparing recombinant vaccinia virus infected TBCLs, the cells were infected with rVac-NP or rVac-pSC11 virus at a multiplicity of infection (moi) of 3. After adsorption for 1 h 30 min at 37 °C, the infected cells were washed and re-suspended with 2% FBS RPMI and incubated overnight prior to functional assay by flow cytometry.

### Intracellular cytokine staining (ICS) by flow cytometry

The flow cytometry assay comprised the FACS tubes of the polyclonal peptide-specific T cell lines (effector cells) stimulated with target cells (TBCLs pulsed with specific peptide or TBCLs infected with rVac-NP virus), and the control tubes including the effector cell alone, effector cells plus unpulsed TBCLs or effector cells plus rVac-pSC11 virus infected TBCLs. Staphylococcal enterotoxin B (SEB) was used as the positive stimulation control. Briefly, $1 \times 10^5$ effector cells were mixed with $1 \times 10^6$ target cells in a one ml volume (effector: target ratio of 1:10). The cell mixtures were incubated with anti-CD107a monoclonal antibody conjugated with FITC (BD Bioscience) for 2 h before adding with the mixture of monensin and brefeldin A at the final concentration of 10 μg/ml, and further incubated for 5 h. The cells were washed with FACS washing buffer (1% FBS, 0.5% NHS, 0.5 mM EDTA, 0.1% $NaN_3$ in PBS) and treated with FACS Permeabilizing Solution II (BD Biosciences) for 15 min in the dark in order to fix and permeabilize the cells. Thereafter, the cells were washed and stained with a cocktail of monoclonal antibodies: anti-CD4 PerCP or anti-CD8 PerCP, anti-IFN-$\gamma$ APC and anti-TNF-$\alpha$ PE monoclonal antibodies (BD Biosciences). The CD4$^+$ or CD8$^+$ T cell populations producing IFN-$\gamma$, TNF-$\alpha$ and upregulation of CD107a degranulation marker were analyzed by FACSCalibur$^{TM}$ instrument by using CellQuest$^{TM}$ program (BD Biosciences).

## Data analysis

The data graphs on IFN-$\gamma$ T cell responses in H5N1 survivors and non-H5N1 subjects were generated using GraphPad Prism software version 4.0. The difference in the magnitude of responses (SFCs/$10^6$ PBMCs) between the H3N2 patients and healthy subjects was analyzed by Mann–Whitney U test. The statistically significant difference was obtained when the $P$-value was less than 0.05.

## RESULTS

### Screening for T cell response against NP, M and HA in H5N1 survivors

Pool peptides derived from H5N1 NP, M or HA amino acid sequences were employed to screen for reactive T cell response in H5N1 survivors using IFN-$\gamma$ ELISpot assay. The results showed that PBMCs from the four H5N1 survivors elicited T cell responses against NP, M and HA peptide pools as shown in Figs. 1A–1C, respectively. The number of IFN-$\gamma$ producing T cells stimulated by NP pooled peptides was higher than those of M and HA, which suggested that NP was more immunogenic in the induction of T cell responses. We also found that the T cell response in adults (survivor nos. 1 and 2) was stronger than that in children (survivor nos. 3 and 4). The T cell reactivity against NP and M peptides persisted for as long as 4 years after onset of disease as followed that far (Figs. 1A and 1B). All four H5N1 survivors had markedly high neutralizing antibody titers against H5N1 virus as demonstrated by an ELISA-based microNT assay. Moreover, they also contained neutralizing antibodies to seasonal H1N1 and/or H3N2 viruses as shown in supplementary Table S1.

### Determining the T cell activity against specific epitopes on NP, M and HA

Due to strong responses to pooled NP, M and HA peptides, H5N1 survivor nos. 1 and 2 were further identified for individual T cell specific epitopes by ELISpot assay. The major epitopes recognized by survivor no. 1 were $NP_{1-20}$, $NP_{111-130}$ (Fig. 2A), $M1_{121-140}$, $M1_{201-220}$ (Fig. 2B) and $HA_{461-480}$ (Fig. 2C); whereas survivor no. 2 recognized $NP_{411-430}$ (Fig. 2D), $M1_{1-20}$, $M1_{91-110}$, $M1_{241-252}$ (Fig. 2E), $HA_{41-60}$, $HA_{251-270}$ and $HA_{291-310}$ (Fig. 2F). However, the magnitude of responses to NP peptides was higher than those to M and HA peptides, indicating that NP is more immunogenic and putatively a highly immunogenic protein target for T cell responses. Based on serial PBMC samples, it was demonstrated that the reactivity of the peptide-reactive T cells could persist for at least 4 years after disease onset. The summary of magnitude and longevity of T cell responses are shown in Table S4.

### Cross -reactive T cell responses against H5N1 NP peptides in non-H5N1 individuals

This study carried out the cross-reactive T cell response to H5N1 NP peptides in 33 non-H5N1 individuals, including PBMCs from convalescent blood samples of 10 H3N2 patients and 23 healthy individuals. These subjects were confirmed for their H5N1 seronegativity by microNT assay (Table S1B). PBMCs were tested against the 49 overlapping peptides spanning the entire H5N1 NP by IFN-$\gamma$ ELISpot assay. The results demonstrated that the magnitude of total cross-reactive T cells varied considerably between individuals. The mean of ex vivo IFN-$\gamma$ responses was 102 and 51 SFCs/$10^6$ PBMCs in H3N2 patients and healthy subjects, respectively; and the mean of responses was statistically significant different between both groups (Mann–Whitney U test; $p < 0.05$) (Fig. 3).

The ELISpot assay with individual candidate NP peptide was conducted to identify the peptide-specific reactivity. From a total of 33 non-H5N1 subjects, six (60%) of 10

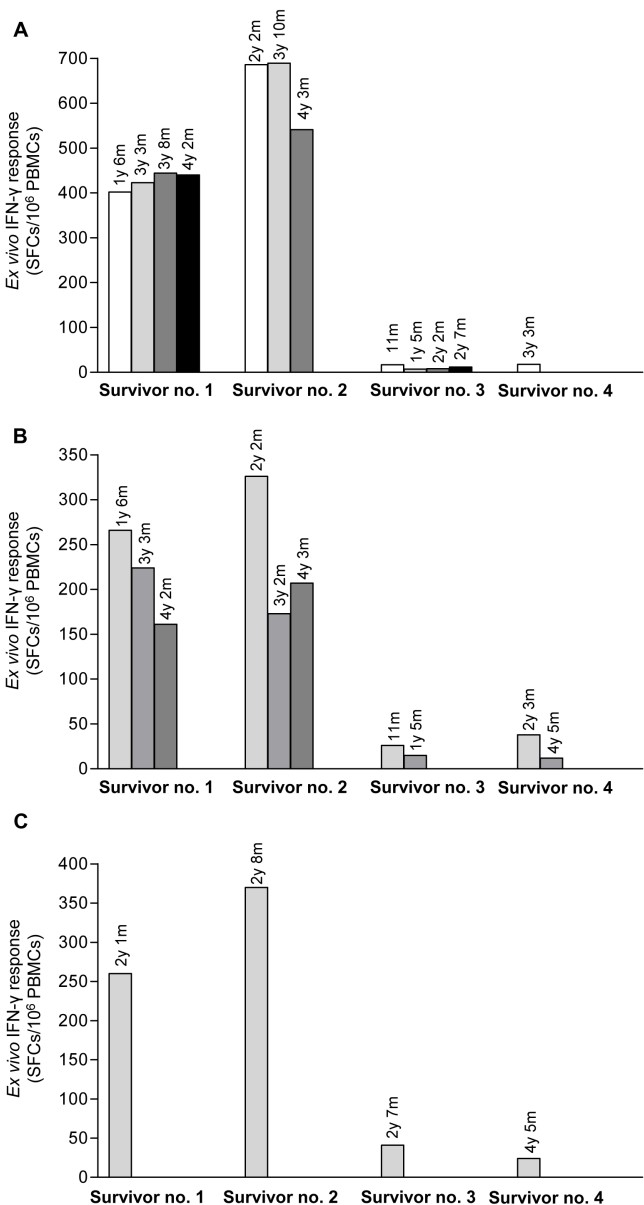

**Figure 1  Total magnitude of T cell responses against H5N1 NP, M and HA peptides in sequential PBMC samples from four H5N1 survivors as measured by IFN-γ ELISpot assay.** The IFN-γ producing T cells against H5N1 overlapping peptides of NP (A), M (B) and HA (C) are indicated. Survivor nos.1 and 2 had higher magnitude of T cell responses against NP and M peptides than survivor nos. 3 and 4.

H3N2 patients and nine (39%) of 23 healthy subjects responded to the stimulation by NP peptides derived from H5N1 KAN-1 virus. In total, 5 peptides were recognized, i.e., $NP_{111-130}$, $NP_{221-230}$, $NP_{311-330}$, $NP_{401-420}$ and $NP_{481-498}$ as indicated in Table 1. However, each responder recognized only 1 or 2 peptides.

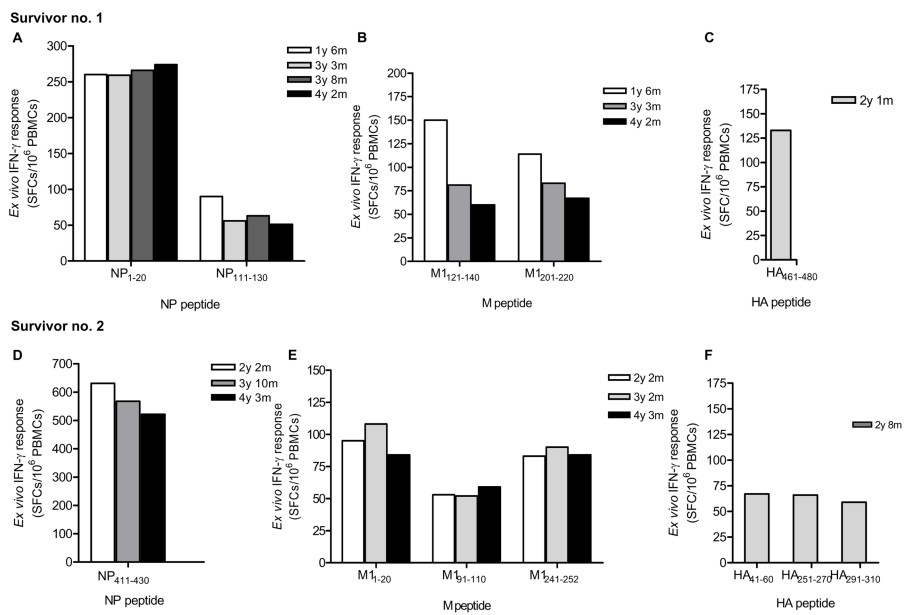

**Figure 2** **Ex vivo IFN-γ responses and longevity of H5N1 virus-specific T cells against NP, M and HA individual peptides in H5N1 survivors.** The responses of peptide-specific T cells of survivor no. 1 (A-C) and survivor no. 2 (D-F) are shown as determined by IFN-γ ELISpot assay.

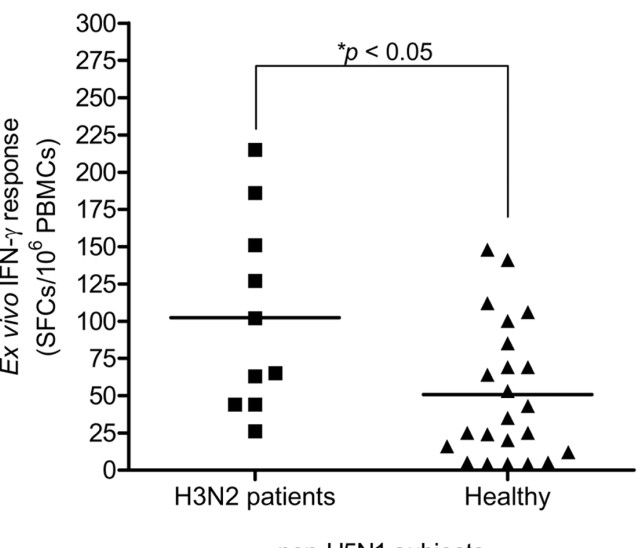

**Figure 3** **Total magnitude of IFN-γ cross-reactive T cell responses in H3N2 patients and healthy individuals.** There was statistically significant difference in mean of magnitude of responses (SFCs/10⁶ PBMCs) between two groups (Mann–Whitney $U$ test $p < 0.05$).

**Table 1  Magnitude of cross-reactive T cell responses to H5N1 NP peptides in non-H5N1 subjects.**

| Subject No. | Peptide | Amino acid sequence | ex vivo IFN-$\gamma$ response (SFCs/$10^6$ PBMCs) |
|---|---|---|---|
| H3N2 patients (six responders from 10 patients, 60%) | | | |
| 2 | $NP_{111-130}$ | YDKEEIRRIWRQANNGEDAT | 40 |
| 6 | $NP_{221-240}$ | RMCNILKGKFQTAAQRAMMD | 140 |
| 7 | $NP_{111-130}$ | YDKEEIRRIWRQANNGEDAT | 60 |
| 8 | $NP_{221-240}$ | RMCNILKGKFQTAAQRAMMD | 62 |
|  | $NP_{311-330}$ | QVFSLIRPNENPAHKSQLVW | 104 |
| 9 | $NP_{221-240}$ | RMCNILKGKFQTAAQRAMMD | 104 |
| 10 | $NP_{311-330}$ | QVFSLIRPNENPAHKSQLVW | 105 |
| Healthy individuals (nine responders from 23 subjects, 39%) | | | |
| 11 | $NP_{481-498}$ | MNNEGSYFFGDNAEEYDN | 36 |
|  | $NP_{311-330}$ | QVFSLIRPNENPAHKSQLVW | 52 |
| 13 | $NP_{221-240}$ | RMCNILKGKFQTAAQRAMMD | 37 |
|  | $NP_{111-130}$ | YDKEEIRRIWRQANNGEDAT | 28 |
| 17 | $NP_{111-130}$ | YDKEEIRRIWRQANNGEDAT | 32 |
| 19 | $NP_{481-498}$ | MNNEGSYFFGDNAEEYDN | 22 |
| 27 | $NP_{111-130}$ | YDKEEIRRIWRQANNGEDAT | 53 |
| 28 | $NP_{481-498}$ | MNNEGSYFFGDNAEEYDN | 28 |
| 29 | $NP_{221-240}$ | RMCNILKGKFQTAAQRAMMD | 71 |
| 30 | $NP_{401-420}$ | ASAGQISVQPTFSVQRNLPF | 45 |
| 31 | $NP_{221-240}$ | RMCNILKGKFQTAAQRAMMD | 37 |
|  | $NP_{401-420}$ | ASAGQISVQPTFSVQRNLPF | 64 |

## H5N1 NP -specific T cells elicited cytokine production and cytotoxic function

The effector cells in bulk culures of peptide-specific T cell lines (generated by clonal expansion of T cells after in vitro stimulation [IVS] of PBMCs from H5N1 survivors no. 1 with $NP_{1-20}$ and survivors no.2 with $NP_{411-430}$ peptide) were investigated by flow cytometry for IFN-$\gamma$ and TNF-$\alpha$ productions, and the upregulation of CD107a, the degranulating marker of cytotoxic function, and additionally their immunophenotypes. The result showed that both $CD4^+$ and $CD8^+$T cells from survivor nos. 1 and 2 expressed IFN-$\gamma^+$ and TNF-$\alpha^+$ in recognition to the target cells infected with rVac-NP virus or pulsed with $NP_{1-20}$ (for survivor no.1) or $NP_{411-430}$ (for survivor no.2). Nevertheless, only $CD8^+$ T cells sufficiently expressed $CD107a^+$, while the expression level was poor for the $CD4^+$T cells (Fig. 4, Fig. S2 and Table S5).

## Identity of NP epitope sequences among different influenza virus subtypes

Amino acid sequences of the H5N1 NP peptides that were recognized by the H5N1 survivors and non-H5N1 subjects were aligned and analyzed for their identities with those derived from the human influenza viruses as shown in Table 2. The H5N1 $NP_{1-20}$ and $NP_{401-420}$ amino acid sequences were identical to those of the H1N1pdm virus, while the $NP_{111-130}$ and $NP_{221-240}$ were identical to the H3N2 virus. As such, the H5N1 cross-reactive T cells

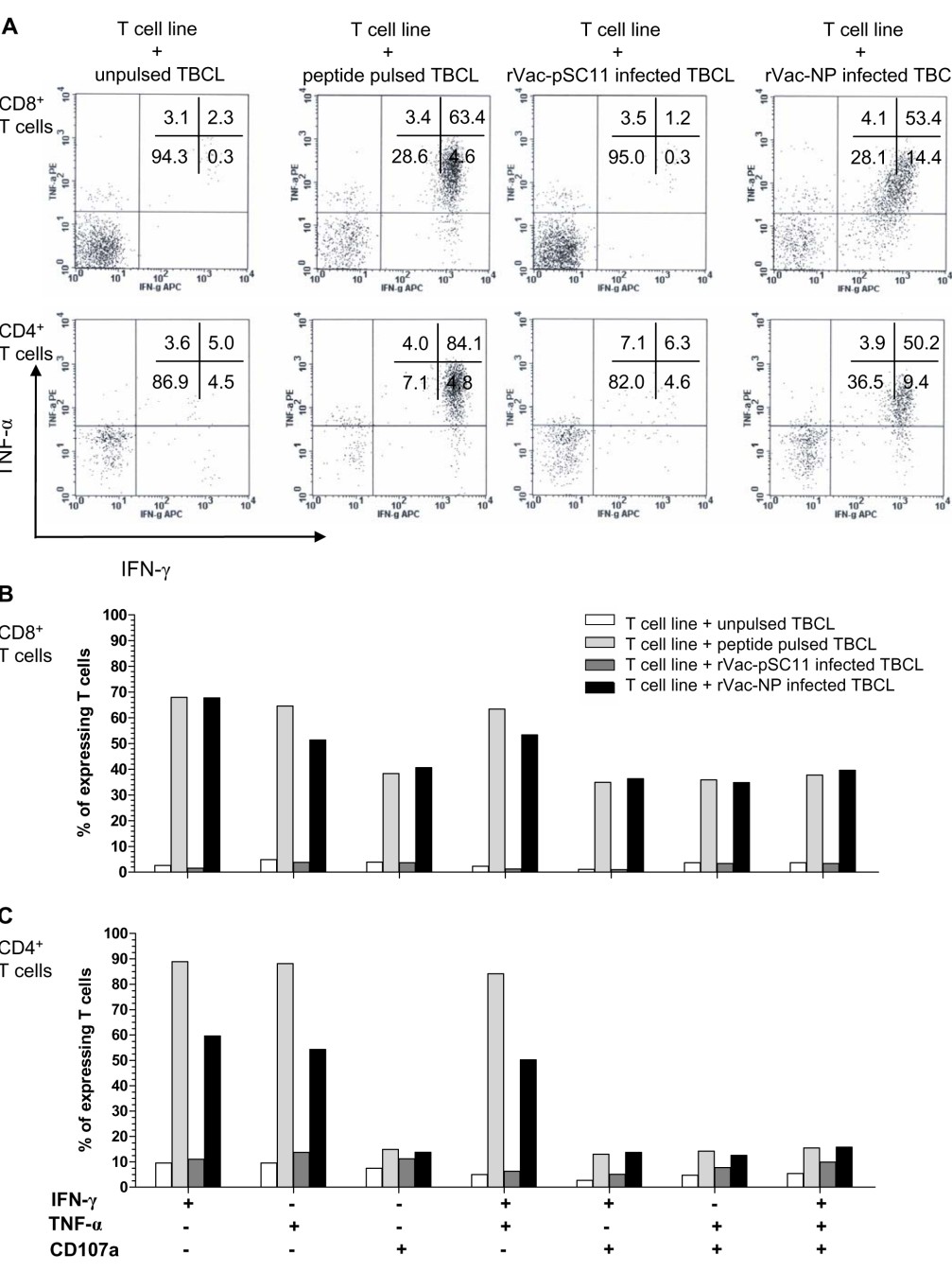

**Figure 4** **Polyfunctional analysis of effector NP$_{411-430}$-specific T cells.** The T cell function which singly or simultaneously produced IFN-$\gamma$ and TNF-$\alpha$ (A) and/or upregulation of CD107a degranulation marker (B and C) in survivor no. 2 are shown. The percentages of both specific CD4$^+$ and CD8$^+$ T cells that expressed IFN-$\gamma$$^+$, TNF-$\alpha$ $^+$ and/or CD107a markedly increased in recognition of the target cells pulsed with peptide or infected with rVac-NP virus.

**Table 2  Amino acid identity and cross reactivity to H5N1 NP peptides.**

| Epitope/ subtype | Sequence | Cross reactivity | Frequency of recognition |
|---|---|---|---|
| NP$_{1-20}$ | | | |
| H5 | MASQGTKRSYEQMETGGERQ | – | 1* survivor |
| H3 | **************D*D** | | |
| H1 | **************D**** | | |
| H1pdm | ******************** | | |
| NP$_{111-130}$ | | | |
| H5 | YDKEEIRRIWRQANNGEDAT | + | 5 (1* survivor) |
| H3 | ******************** | | |
| H1 | ***************D*** | | |
| H1pdm | *****V************** | | |
| NP$_{221-240}$ | | | |
| H5 | RMCNILKGKFQTAAQRAMMD | + | 6 |
| H3 | ******************** | | |
| H1 | **************K**** | | |
| H1pdm | ******************V* | | |
| NP$_{311-330}$ | | | |
| H5 | QVFSLIRPNENPAHKSQLVW | + | 3 |
| H3 | *IY***************** | | |
| H1 | *VY***************** | | |
| H1pdm | *****M************** | | |
| NP$_{401-420}$ | | | |
| H5 | ASAGQISVQPTFSVQRNLPF | + | 2 |
| H3 | *****T************** | | |
| H1 | *******T*********** | | |
| H1pdm | ******************** | | |
| NP$_{411-430}$ | | | |
| H5 | TFSVQRNLPFERATIMAAFT | – | 1* survivor |
| H3 | **********KS****** | | |
| H1 | *********DKT****** | | |
| H1pdm | ******************S | | |
| NP$_{481-498}$ | | | |
| H5 | MNNEGSYFFGDNAEEYDN | + | 3 |
| H3 | *S*************** | | |
| H1 | *S*************** | | |
| H1pdm | *S*************S | | |

**Notes.**

*H5N1 survivor subject.

NP from H5N1 strain A/Thailand/1(KAN-1)/2004 (H5N1) (accession no. AAV35112); H3N2 strain A/Brisbane/10/2007 (H3N2) (accession no. ACO95274); H1N1 strain A/New Caledonia/20/1999 (H1N1) (accession no. ABW81621); H1N1pdm strain A/Thailand/104/2009 (H1N1) (accession no. ACR23306).

- Cross reactivity: (–) epitope recognized by H5N1 survivor only; (+) epitope recognition by H5N1 survivor and/or non-H5N1 subjects.

found in non-H5N1 subjects ($NP_{111-130}$, $NP_{221-240}$, $NP_{311-330}$, $NP_{401-420}$, and $NP_{481-498}$ peptides) might be generated from previous exposure to the common epitopes between the H5N1 virus and the seasonal H1N1 or H3N2 viruses. Cross reactivity for the $NP_{111-130}$ and $NP_{221-240}$ was found in higher frequency than the other peptides. Interestingly, the $NP_{411-430}$ (TFSVQRNLPFERATIMAAFT) was a unique epitope recognized by H5N1 survivors only.

## DISCUSSION

Thailand reported 25 H5N1 infected cases with 17 deaths (fatality rate 68%) to WHO during the outbreak that lasted from 2004 to 2006. No H5N1 patient has occurred since then. We could access four of the eight survivors, whose sequential blood samples were collected at approximately six-month intervals. Using cryopreserved PBMC samples from four H5N1 survivors and 33 non-H5N1 subjects, the present study employed IFN-$\gamma$ ELISpot assay and flow cytometry to investigate cell-mediated immune responses targeting NP, M and HA derived from HPAI H5N1 virus strain A/Thailand/1(KAN-1)/2004 (H5N1) (*Puthavathana et al., 2005*; *Kitphati et al., 2009*). Our analyses on the amino acid sequence identities of various H5N1 viruses isolated in Thailand between 2004 and 2008 against the KAN-1 virus showed that the peptides used were relatively conserved across the circulating viruses during the study period. NP was the most conserved protein with the degree of identities varying from 99.1–100%, whereas the identities among M1, M2, and HA proteins ranging from 98.8–100%, 96.9–100%, and 98.7–99.8%, respectively. The study demonstrated that adult survivors (nos. 1 and 2) had stronger T cell response than the children survivors (nos. 3 and 4) which could be explained by few reasons: (1) adults were previously expose to the seasonal influenza virus infection or vaccination; therefore, the pre-existing cross-reactive T cells might be boosted after H5N1 infection, and also adding up with more number of H5N1-specific T cells; (2) young children generated a lower number of influenza virus-specific memory T cells compared to adults; or the memory T cells from children had shorter half-life; and (3) young children (survivor no. 3) who developed mild disease after H5N1 virus infection might generate a lower number of specific memory T cells than adult survivors who developed severe disease. It has been previously reported that the 2009 H1N1pdm patients who developed severe disease elicited higher levels of circulating influenza virus-specific $CD4^+$ T cells to NP and M when compared to the cases with mild disease as measured by ELISpot assay (*Zhao et al., 2012*).

The total number of IFN-$\gamma$ reactive T cells against NP was higher than that of M and HA as demonstrated in two H5N1 adult survivors by ELISpot assay using 20 mers-overlapping pooled peptides, which suggested that NP was more immunogenic in the induction of T cell response. NP is the most abundant viral protein synthesized in the infected cells. It contains several immunodominant epitopes that could stimulate both the humoral and cell-mediated immune responses. As such, NP was identified as the major target of subtype-specific and cross-subtypic $CD4^+$ and cytotoxic $CD8^+$ T lymphocytes as reported by several groups of previous investigators (*Kreijtz et al., 2008*; *Lee et al., 2008*; *Rimmelzwaan & Katz, 2013*; *Roti et al., 2008*). Our result of the 1st round ELISpot assay

using pooled peptides was confirmed by the 2nd round ELISpot assay using individual peptides. T cell responses could last longer than 4 years after disease onset for as far as the PBMC samples were available. Using individual peptide, our results showed that each survivor recognized different peptides, suggesting that their HLA or their repertoire of specific T cells were different.

Up until the present day, the H5N1-specific T cell epitopes are not well characterized due to the small number of H5N1 survivors. *Powell et al. (2012)* could identify three H5 HA-specific T cell epitopes ($HA_{160-177}$, $HA_{344-364}$ and $HA_{439-456}$) from H5N1 asymptomatic cases; while *Sun et al. (2010)* could identify only one H5 HA-specific T cell epitope ($HA_{205-214}$). However, our study was confined to the H5N1 NP peptides according to the limited amount of PBMCs available and the high immunogenicity of NP epitopes. Amino acid sequence identity of greater than 90% was found across NP proteins of various influenza virus subtypes (*Noisumdaeng et al., 2014*). Among 49 NP overlapping peptides, three peptides ($NP_{1-20}$, $NP_{111-130}$ and $NP_{411-430}$) were identified among the four H5N1 survivors, and five peptides ($NP_{111-130}$, $NP_{221-230}$, $NP_{311-330}$, $NP_{401-420}$ and $NP_{481-498}$) among 33 non-H5N1 subjects. Interestingly, only one peptide, $NP_{111-130}$, could be identified in both H5N1 survivors and non-H5N1 subjects. Based on A/Viet Nam/CL26/2004 (H5N1) sequence, the study in healthy Vietnamese and English subjects demonstrated the cross-reactive $CD4^+$T cells against H5N1 $NP_{221-238}$, $NP_{404-420}$ and $NP_{478-493}$ as determined by ELISpot assay (*Lee et al., 2008*). Moreover, the cross-reactive $CD4^+$T cells against H5N1 $NP_{401-420}$ and $NP_{113-132}$ were demonstrated in the healthy Caucasian descent based on A/Vietnam/1203/2004 H5N1 (VN1203) as determined by tetramer-guide epitope mapping. The $NP_{401-420}$ was restricted to HLA-DR0404; while $NP_{113-132}$ was restricted to HLA-DR1101 (*Roti et al., 2008*). However, the NP epitope variants which could escape from the T cell recognition have been reported (*Rimmelzwaan et al., 2004*; *Berkhoff et al., 2007*).

We further investigated the effector functions and immunophenotypes of the NP peptide-specific T cells by flow cytometry. The $NP_{1-20}$ and $NP_{411-430}$ peptides could stimulate both $CD4^+$ and $CD8^+$ T cell subsets in survivors, leading to clonal expansion and a high degree of polyfunction by simultaneously producing IFN-$\gamma$ and TNF-$\alpha$, and together with an upregulation of CD107a in recognition of the target cells pulsed with peptide or infected with rVac-NP virus. Since our peptides are 20 amino acids long; they might bind to both HLA class I (optimal epitope are 8–12 amino acids long) and class II (optimal epitope are 12–18 amino acids long) and lead to activation of $CD4^+$ and $CD8^+$ memory T cells. It is implied that NP protein expressed in the rVac-NP virus infected TBCLs is naturally processed and presented in association with HLA to the effector T cells (*Jameson, Cruz & Ennis, 1998*). Importantly, it is uncertain that the induction of such polyfunctional T cell populations might be associated with the recovery from severe diseases in our H5N1 survivors. The amino acid sequences of the avian H5N1 internal proteins were closely identical to those of the seasonal H1N1, H3N2 and H1N1pdm viruses; exposure to seasonal influenza virus infection or vaccination can generate a pool of cross-reactive memory $CD4^+$ and $CD8^+$ T cells which are capable of recognizing a number of conserved internal proteins form avian H5N1 virus.

## CONCLUSIONS

The broader vaccines that rely on the induction of T cell-based immunity against conserved epitopes would provide broader partial protection, restrict the viral diversity in the infected host and help lower severity and mortality against overwhelming pandemic influenza virus infection. Our present study provides insight into a better understanding of the homosubtypic and heterosubtypic T cell-mediated immune responses against H5N1 virus in H5N1 survivors and non-H5N1 subjects. NP is an immunodominant target of cross-recognition owing to its high conservancy. Therefore, the development of vaccine targeting the conserved NP may be a novel strategy for influenza vaccine design.

## ACKNOWLEDGEMENTS

We would like to thank all subjects who participated in the study. We thank all medical staff from Department of Disease Control, Ministry of Public Health for their help in specimen collection from H5N1 survivors and research assistants at Siriraj Influenza Cooperative Research Center, Department of Microbiology, Faculty of Medicine Siriraj Hospital, Mahidol University for kindly supporting reagents, materials and techniques. P.N. specially thanks Thammasat University Research Unit in Modern Microbiology and Public Health Genomics, Thammasat University.

### Funding

This work was supported by the Thailand Research Fund for Senior Research Scholar through Prof. Pilaipan Puthavathana, the National Science and Technology Development Agency, the Office of the Higher Education Commission and Mahidol University under the National Research Universities Initiative, and Thammasat University Research Unit in Modern Microbiology and Public Health Genomics, Thammasat University. The funders had no role in study design, data collection and analysis, decision to publish, or preparation of the manuscript.

### Grant Disclosures

The following grant information was disclosed by the authors:
Thailand Research Fund for Senior Research Scholar.
National Science and Technology Development Agency.
National Research Universities Initiative.
Thammasat University Research Unit in Modern Microbiology and Public Health Genomics, Thammasat University.

### Competing Interests

The authors declare there are no competing interests.

## Author Contributions

- Pirom Noisumdaeng and Thaneeya Roytrakul conceived and designed the experiments, performed the experiments, analyzed the data, prepared figures and/or tables, authored or reviewed drafts of the paper, and approved the final draft.
- Jarunee Prasertsopon, Phisanu Pooruk and Hatairat Lerdsamran performed the experiments, analyzed the data, prepared figures and/or tables, and approved the final draft.
- Susan Assanasen, Rungrueng Kitphati and Prasert Auewarakul analyzed the data, authored or reviewed drafts of the paper, provided the clinical specimens, and approved the final draft.
- Pilaipan Puthavathana conceived and designed the experiments, analyzed the data, prepared figures and/or tables, authored or reviewed drafts of the paper, and approved the final draft.

## Human Ethics

The following information was supplied relating to ethical approvals (i.e., approving body and any reference numbers):

This study was approved by Institutional Review Boards from the Faculty of Medicine Siriraj Hospital, Mahidol University, under the approval number Si213/2005.

## Data Availability

The raw data are available in the Supplementary Files.

## Supplemental Information

Supplemental information for this article can be found online at http://dx.doi.org/10.7717/peerj.11021#supplemental-information.

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
