# Peer review of "T cell mediated immunity against influenza H5N1 nucleoprotein, matrix and hemagglutinin derived epitopes in H5N1 survivors and non-H5N1 subjects"

_PeerJ, doi:10.7717/peerj.11021_

## Round 0.1 · original submission · Minor Revisions

Dear Dr. Noisumdaeng and colleagues:

Thanks for submitting your manuscript to PeerJ. I have now received two independent reviews of your work, and as you will see, the reviewers raised some minor concerns about the research. Thus, these reviewers are optimistic about your work and the potential impact it will have on research studying T cell mediated immunity against influenza viruses. Thus, I encourage you to revise your manuscript, accordingly, taking into account all of the concerns raised by both reviewers.

In your revision, please provide the missing information noted by the reviewers, and also elaborate on the issues raised by reviewer 2. Finally, please cull any redundancy with tables and figures.

I look forward to seeing your revision, and thanks again for submitting your work to PeerJ.

Best,

-joe

Reviewer 1 ·

Basic reporting

The paper is clearly written and well organized. The introduction and background are well justified.
Tables and figures are comprehensive and helpful to understand the manuscript.

Experimental design

The experiments were designed nicely.
Please mention the data analysis software/tests to generate the figures in the "materials and methods" section.

Validity of the findings

No comments

Reviewer 2 ·

Basic reporting

No comment

Experimental design

No comment

Validity of the findings

No comment

Additional comments

The study determined the T cell responses against the H5N1 HPAI virus in four survivors and 33 non-H5N1 subjects including 10 H3N2 patients and 23 healthy individuals using IFN-γ ELISpot and microneutralization assays and flow cytometry. The study reported profound T cell responses mostly against selected H5N1 NP epitopes in PBMCs of two adult survivors which lasted over 4 years post infection. The study tested an array of H5N1-specific T cell epitopes located in HA, NA and NP peptides and came out with few unique peptides in H5N1 and non-H5N1 survivors. In addition, 60% of the H3N2 patients showed cross-reactive T cell responses against H5N1 NP peptides owing to sequence homology between these viruses.
The data on T cell responses in H5N1 human survivors are valuable. Identification of unique and conserved epitopes in NP gene showing high immunogenicity would guide in designing T cell based vaccines for broad subtype viral protection. The manuscript is well written with clear methodology.
However, there are few issues that authors need to discuss in the manuscript. For example:
a. Size of the cohort: The study involves a small (only four H5N1 subjects) patient cohort from 2004.
b. It was not mentioned in the text which clade of the H5N1 virus was detected in the four H5N1 survivors.
c. Peptides generated from H5N1 virus of clade 1 was used for T cells response analysis. It is necessary to discuss how the T cell responses data of the study would correlate with the T cell responses against currently circulating H5N1 viruses in Thailand?
d. The reason for higher immunomodulatory property of NP peptides needs discussion.
e. Repetition of data should be avoided: Figure 2 and Table 1; and Figure 3 and Table 2.

---

## Round 0.2 · accepted · Accept

Dear Dr. Noisumdaeng and colleagues:

Thanks for revising your manuscript based on the concerns raised by the reviewers. I now believe that your manuscript is suitable for publication. Congratulations! I look forward to seeing this work in print, and I anticipate it being an important resource for groups studying T cell mediated immunity against influenza viruses. Thanks again for choosing PeerJ to publish such important work.

Best,

-joe